# Pulmonary vascular dysfunction among people aged over 65 years in the community in the Atherosclerosis Risk In Communities (ARIC) Study: A cross-sectional analysis

Kanako Teramoto[1,2☯], Mário Santos[1,3,4☯], Brian Claggett[1], Jenine E. John[1], Scott D. Solomon[1], Dalane Kitzman[5], Aaron R. Folsom[6], Mary Cushman[7,8], Kunihiro Matsushita[9], Hicham Skali[1], Amil M. Shah[1]*

1 Division of Cardiovascular Medicine, Brigham and Women's Hospital, Boston, Massachusetts, United States of America, 2 Division of Cardiology, St. Marianna University School of Medicine Hospital, Kawasaki, Japan, 3 Department of Physiology and Cardiothoracic Surgery, Cardiovascular R&D Unit, Faculty of Medicine, University of Porto, Portugal, 4 Department of Cardiology, Hospital Santo António, Porto Hospital Center, Porto, Portugal, 5 Wake Forest University School of Medicine, Winston-Salem, North Carolina, United States of America, 6 Division of Epidemiology and Community Health, School of Public Health, University of Minnesota, Minneapolis, Minnesota, United States of America, 7 Department of Medicine, University of Vermont, Burlington, Vermont, United States of America, 8 Department of Pathology, University of Vermont, Burlington, Vermont, United States of America, 9 Department of Epidemiology, Johns Hopkins Bloomberg School of Public Health, Baltimore, Maryland, United States of America

☯ These authors contributed equally to this work.
* ashah11@rics.bwh.harvard.edu

**Data Availability Statement:** The ARIC datasets are available through BioLINCC, with appropriate

## Abstract

### Background

Heart failure (HF) risk is highest in late life, and impaired pulmonary vascular function is a risk factor for HF development. However, data regarding the contributors to and prognostic importance of pulmonary vascular dysfunction among HF-free elders in the community are limited and largely restricted to pulmonary hypertension. Our objective was to define the prevalence and correlates of abnormal pulmonary pressure, resistance, and compliance and their association with incident HF and HF phenotype (left ventricular [LV] ejection fraction [LVEF] ≥ or < 50%) independent of LV structure and function.

### Methods and findings

We performed cross-sectional and time-to-event analyses in a prospective epidemiologic cohort study, the Atherosclerosis Risk in Communities study. This is an ongoing, observational study that recruited 15,792 persons aged 45–64 years between 1987 and 1989 (visit 1) from four representative communities in the United States: Minneapolis, Minnesota; Jackson, Mississippi; Hagerstown, Maryland; and Forsyth County, North Carolina. The current analysis included 2,810 individuals aged 66–90 years, free of HF, who underwent echocardiography at the fifth study visit (June 8, 2011, to August 28, 2013) and had measurable tricuspid regurgitation by spectral Doppler. Echocardiography-derived pulmonary artery systolic pressure (PASP), pulmonary vascular resistance (PVR), and pulmonary arterial

study approvals consistent with NIH policies. Data request forms through BioLINCC can be accessed at https://biolincc.nhlbi.nih.gov/studies/aric/.

**Funding:** The Atherosclerosis Risk in Communities Study is carried out as a collaborative study supported by National Heart, Lung, and Blood Institute contracts (HHSN268201100005C, HHSN268201100006C, HHSN268201100007C, HHSN268201100008C, HHSN268201100009C, HHSN268201100010C, HHSN268201100011C, and HHSN268 201100012C). The Longitudinal Investigation of Thromboembolism Etiology study was funded by grant R01HL59367. The work for this manuscript was also supported by NHLBI grants R01HL135008, R01HL143224, R01HL150342, R01HL148218, and K24HL152008 (AMS) and a Watkins Discovery Award from the Brigham and Women's Heart and Vascular Center (AMS). The funders had no role in study design, data collection and analysis, decision to publish, or preparation of the manuscript.

**Competing interests:** I have read the journal's policy and the authors of this manuscript have the following competing interests: AMS reports receiving research support from Novartis through the Brigham and Women's Hospital and consulting fees from Bellerophon and Philips Ultrasound. SDS reports receiving research grants from Alnylam, Amgen, AstraZeneca, Bellerophon, Bayer, BMS, Celladon, Cytokinetics, Eidos, Gilead, GSK, Ionis, Lone Star Heart, Mesoblast, MyoKardia, NIH/ NHLBI, Novartis, Sanofi Pasteur, and Theracos and has consulted for Akros, Alnylam, Amgen, Arena, AstraZeneca, Bayer, BMS, Cardior, Cardurion, Corvia, Cytokinetics, Daiichi-Sankyo, Gilead, GSK, Ironwood, Merck, Myokardia, Novartis, Roche, Takeda, Theracos, Quantum Genetics, Cardurion, AoBiome, Janssen, Cardiac Dimensions, Tenaya, Sanofi-Pasteur, Dinaqor, and Tremeau. The remaining authors report no competing interests.

**Abbreviations:** ARIC, Atherosclerosis Risk in Communities; BMI, body mass index; CAD, coronary artery disease; CHD, coronary heart disease; CKD, chronic kidney disease; COPD, chronic obstructive pulmonary disease; DBP, diastolic blood pressure; DPAP, diastolic pulmonary artery pressure; ESC HFA, European Society of Cardiology Heart Failure Association; $FEV_1$, forced expiratory volume in 1 second; FVC, forced vital capacity; HF, heart failure; HFpEF, HF with preserved LVEF; HFrEF, HF with reduced LVEF; IRB, institutional review board; LAVi, left atrial volume index; LV, left ventricular; LVEDD, LV end-diastolic diameter; LVEF, LV ejection fraction; LVMi, LV mass index; MI, myocardial infarction; MPAP, mean pulmonary artery pressure; MWT,

compliance (PAC) were measured. The main outcome was incident HF after visit 5, and key secondary end points were incident HF with preserved LVEF (HFpEF) and incident HF with reduced LVEF (HFrEF). The mean ± SD age was 76 ± 5 years, 66% were female, and 21% were black. Mean values of PASP, PVR, and PAC were 28 ± 5 mm Hg, 1.7 ± 0.4 Wood unit, and 3.4 ± 1.0 mL/mm Hg, respectively, and were abnormal in 18%, 12%, and 14%, respectively, using limits defined from the 10th and 90th percentile limits in 253 low-risk participants free of cardiovascular disease or risk factors. Left heart dysfunction was associated with abnormal PASP and PAC, whereas a restrictive ventilatory deficit was associated with abnormalities of PASP, PVR, and PAC. PASP, PVR, and PAC were each predictive of incident HF or death (hazard ratio per SD 1.3 [95% CI 1.1–1.4], $p < 0.001$; 1.1 [1.0–1.2], $p = 0.04$; 1.2 [1.1–1.4], $p = 0.001$, respectively) independent of LV measures. Elevated pulmonary pressure was predictive of incident HFpEF (HFpEF: 2.4 [1.4–4.0, $p = 0.001$]) but not HFrEF (1.4 [0.8–2.5, $p = 0.31$]). Abnormal PAC predicted HFrEF (HFpEF: 2.0 [1.0–4.0, $p = 0.05$], HFrEF: 2.8 [1.4–5.5, $p = 0.003$]), whereas abnormal PVR was not predictive of either (HFpEF: 0.9 [0.4–2.0, $p = 0.85$], HFrEF: 0.7 [0.3–1.4, $p = 0.30$],). A greater number of abnormal pulmonary vascular measures was associated with greater risk of incident HF. Major limitations include the use of echo Doppler to estimate pulmonary hemodynamic measures, which may lead to misclassification; inclusions bias related to detectable tricuspid regurgitation, which may limit generalizability of our findings; and survivor bias related to the cohort age, which may result in underestimation of the described associations.

## Conclusions

In this study, we observed abnormalities of PASP, PVR, and PAC in 12%–18% of elders in the community. Higher PASP and lower PAC were independently predictive of incident HF. Abnormally high PASP predicted incident HFpEF but not HFrEF. These findings suggest that impairments in pulmonary vascular function may precede clinical HF and that a comprehensive pulmonary hemodynamic evaluation may identify pulmonary vascular phenotypes that differentially predict HF phenotypes.

## Author summary

### Why was this study done?

- Abnormal pressure or function of the blood vessels in the lungs is associated with risk of developing heart failure. These blood vessels in the lungs can be characterized by pressure, resistance, and compliance.

- Alterations in heart and lung function can influence the pulmonary blood vessels.

- The risk of heart failure is highest in late life. However, little is known about the association of heart and lung dysfunction with pulmonary vascular measures, and the association of pulmonary vascular measures with heart failure risk, among elders in the community.

mean wall thickness; NT-proBNP, N-terminal prohormone brain natriuretic peptide; PAC, pulmonary arterial compliance; PAD, peripheral artery disease; PAR, population attributable risk; PASP, pulmonary artery systolic pressure; PH, pulmonary hypertension; PVR, pulmonary vascular resistance; RC time, resistance-compliance product; RV, right ventricular; RVFAC, RV fractional area change; RWT, relative wall thickness; STROBE, Strengthening the Reporting of Observational Studies in Epidemiology; SV, stroke volume; TA S', tricuspid annular peak systolic velocity; TR, tricuspid regurgitation; VTE, venous thromboembolism; VTI, time-velocity integral; WU, Wood unit.

## What did the researchers do and find?

- We performed a cross-sectional analysis and an analysis of associations with subsequent occurrence of clinical heart failure in a large community-based cohort with participants aged 66–90 years.

- We measured the cross-sectional association of left heart disease, pulmonary dysfunction, and venous thromboembolism with abnormalities in these three major pulmonary vasculature measures (i.e., pulmonary pressure, resistance, and compliance).

- The presence of left heart disease was associated with abnormally higher pulmonary pressure. Pulmonary dysfunction was associated with abnormalities in all three pulmonary vasculature measures (i.e., pressure, resistance, and compliance).

- We performed time-to-event analysis to investigate the association of abnormalities in pulmonary vasculature measures with future development of heart failure and heart failure subtypes (heart failure with preserved or reduced left ventricular ejection fraction).

- We found that higher pulmonary pressure and lower compliance were associated with risk of heart failure. Higher pulmonary pressure was associated with greater risk of heart failure with preserved ejection fraction but not with reduced ejection fraction. Lower pulmonary arterial compliance was more strongly associated with heart failure with reduced ejection fraction.

## What do these findings mean?

- These findings suggest that abnormalities of pulmonary pressure and compliance may precede the development of heart failure and are associated with increased risk for heart failure.

- Further research is needed to better understand whether, and how, these abnormalities in the pulmonary vasculature directly contribute to heart failure development and whether this differs by heart failure subtype.

- The main limitations of this study include potential error in pulmonary vascular measures due to the methodology used and the selection of participants in this analysis, which may limit the generalizability of these findings.

## Introduction

Pulmonary vascular disease predicts worse outcomes in cardiovascular and pulmonary diseases [1] and is most commonly secondary to left heart, lung, venous thromboembolic, and primary pulmonary vascular diseases [2]. Although most commonly manifest clinically as pulmonary hypertension (PH), the pulmonary circulation can also be characterized in terms of pulmonary vascular resistance (PVR) and pulmonary arterial compliance (PAC), which are inversely related measures of hydraulic load [3]. In addition to elevated left ventricular (LV) filling pressure, higher pulmonary pressures may result from reciprocal changes in PVR and PAC. Indeed, recent autopsy-based data demonstrate evidence of global pulmonary vascular remodeling in patients with heart failure (HF) and PH [4]. Importantly, these pulmonary

hemodynamic measures are not redundant, and pulmonary pressure may remain normal despite impairments in PVR and PAC [5]. Pulmonary pressure and vascular resistance increase with age [6,7], in parallel with age-associated changes in LV compliance and filling pressure, pulmonary function, and vascular function [8,9]. Although prior studies have described the association of PASP with incident HF, few data are available regarding the prevalence, causes, and prognostic implications of PH and vascular dysfunction in the elderly.

Invasive right heart catheterization is the gold-standard method to assess the pulmonary vasculature but cannot be broadly applied owing to its invasive nature. Doppler echocardiography is a reliable, validated, and widely used method to assess pulmonary hemodynamics, including pulmonary artery systolic pressure (PASP), mean pulmonary artery pressure (MPAP), and PVR [10,11]. We aimed to determine the prevalence and determinants of pulmonary vascular dysfunction and define their prognostic relevance for incident HF with preserved LV ejection fraction (LVEF) (HFpEF) and HF with reduced LVEF (HFrEF) in late life. We studied community-based participants aged 66–90 years in the Atherosclerosis Risk in Communities (ARIC) study who underwent comprehensive transthoracic echocardiography at the fifth study visit.

## Methods

### Study population

The ARIC study is an ongoing, prospective observational cohort study [12]. ARIC recruited 15,792 persons aged 45–64 years between 1987 and 1989 (visit 1) from four communities in the United States: Forsyth County, North Carolina; Jackson, Mississippi; suburban Minneapolis, Minnesota; and Washington County, Maryland. Between June 2, 2011, and August 30, 2013, 6,538 participants returned for a fifth visit that included anthropometrics, interviewer-administered questionnaires (https://sites.cscc.unc.edu/aric/cohort-forms-forms), laboratory testing, and comprehensive echocardiography. The current analysis included 2,810 visit 5 participants free of prevalent HF and with measurable tricuspid regurgitation (TR) spectral Doppler on echocardiography. Prevalent HF at visit 5 was defined as an adjudicated HF hospitalization between 2005 and visit 5 (HF adjudication in ARIC began in 2005), hospitalization with an HF ICD code 428 in any position prior to 2005 [13,14], or any self-report of HF or HF medication use on an annual ARIC follow-up phone interviews prior to visit 5. Prospective protocols for the ARIC study overall and echocardiography at visit 5 have been previously published [15]. This analysis did not have a specific prospective protocol.

### Ethics statement

The ARIC study has been approved by institutional review boards (IRBs) at all participating institutions. All participants provided written informed consent at all study visits.

### Assessment of pulmonary hemodynamics

Studies were acquired in all participants attending visit 5 at all four field centers by certified study sonographers using uniform imaging equipment and following a standardized image acquisition protocol as previously described [15]. Quantitative measures were performed by blinded analysts at a dedicated core laboratory based on the recommendations of the American Society of Echocardiography [16,17]. Reproducibility metrics have been previously reported [15] and demonstrated intrareader coefficients of variation of less than 15% and intraclass correlations of greater than 0.84. PASP was calculated as $4^*(\text{peak velocity}_{TR}^2) + 5$ [11,18,19]. MPAP was estimated as the mean systolic right ventricle–to–right atrium pressure

gradient by tracing the TR time-velocity integral (VTI) as previously validated [11,20]. Diastolic pulmonary artery pressure (DPAP) was calculated using the formula: MPAP = 1/3 PASP + 2/3 DPAP. Stroke volume (SV) was calculated using the LV outflow tract diameter and VTI. PVR was calculated as $10^*$TR velocity/TVI$_{RVOT}$ [21]. PAC was calculated as SV/pulmonary pulse pressure and is expressed in mL/mm Hg [22].

Age-appropriate reference limits for pulmonary vascular measures were defined in a subset of low-risk participants free of prevalent cardiovascular disease or risk factors (referred to as the "low-risk reference subgroup"). This group excluded participants with prevalent CV disease defined as coronary heart disease (CHD; includes prior myocardial infarction [MI] or coronary intervention, or regional wall motion abnormality on echocardiography), prior HF hospitalization or HF self-report, atrial fibrillation, and moderate or greater valvular disease at visit 5; or CV risk factors including hypertension, diabetes, body mass index (BMI) of >30 or <18.5 kg/m$^2$, chronic kidney disease (CKD) defined as an eGFR <60 ml/min/1.73 m$^2$, QRS duration ≥120 milliseconds, or active smoking.

## Assessment of potential causes of pulmonary vascular dysfunction

Left heart dysfunction was defined as an LVEF <50% or echocardiographic evidence of elevated left atrial pressure based on left atrial enlargement (left atrial volume index [LAVi] > 34 ml/m$^2$) or elevated E/e' ratio (average E/e' > 14) [17] on visit 5 echocardiogram. Pulmonary dysfunction was defined based on participant self-report and visit 5 spirometry. A restrictive pattern was defined as a percent of predicted forced vital capacity (FVC) <80% [23,24]. Chronic obstructive pulmonary disease (COPD) was defined based on self-report of physician diagnosis or an obstructive spirometric pattern defined as a forced expiratory volume in 1 second (FEV$_1$)/FVC ratio <0.70 [25]. History of prior venous thromboembolism (VTE; includes deep vein thrombosis or pulmonary embolus) was assessed through physician adjudication of participant hospitalizations with a broad set of VTE ICD-9-CM discharge codes as previously described [26].

## Clinical covariates

Prevalent hypertension and diabetes were based on blood pressure and glucose measured from study visits 1 through 5, self-report of physician diagnosis, and medication use as previously described [12]. Detailed interview data collection forms are available at the ARIC website (https://sites.cscc.unc.edu/aric/desc_pub). Obesity was defined as a BMI >30 kg/m$^2$. CKD was defined as an eGFR <60 mL/min/1.73 m$^2$ calculated using the CKD-EPI equation [27]. Prevalent CHD was based on ARIC surveillance for CHD events including definite or probable MI and coronary revascularization [28]. Atrial fibrillation was ascertained based on ECGs at five study visits and hospital discharge records [29].

## Assessment of right ventricular function, biomarkers, and clinical outcomes

Right ventricular (RV) function was assessed on visit 5 echocardiograms as both (1) the RV fractional area change (RVFAC), calculated as $100^*$(end-diastolic area − end-systolic area)/end-diastolic area, where RV areas were measured in an RV-focused apical 4-chamber view; and (2) tissue Doppler-based tricuspid annular peak systolic velocity (TA S') [30]. Hs-TnT was measured using a highly sensitive assay (Elecsys Troponin T, Roche Diagnostics, Indianapolis, IN). N-terminal prohormone brain natriuretic peptide (NT-proBNP) was measured using electrochemiluminescent immunoassay (Roche Diagnostics) with a lower detection limit of ≤5 ng/mL [31].

Incident HF post visit 5 was ascertained based on physician adjudication aided by comprehensive abstraction of medical records from hospitalizations with an ICD code related to potential HF as previously described [13]. Abstracted items included measures of LVEF (S1 Text). Incident HFpEF was defined as an adjudicated HF event with a documented LVEF ≥50%, whereas incident HFrEF was defined if the LVEF was <50%. Deaths were ascertained by ARIC surveillance or the National Death Index [28]. For this analysis, all participants were followed through December 31, 2018.

## Statistical methods

Echocardiographic measures of pulmonary hemodynamics were described in the low-risk reference subgroup overall and stratified by sex using quantile regression (STATA qreg) to define the 10th and 90th percentile limits with associated 95% confidence limits. Abnormal PASP and PVR were defined as values above the resulting 90th percentile limits, whereas abnormal PAC was defined as values below the 10th percentile limit. Among ARIC participants attending visit 5 and free of prevalent HF, the cross-sectional associations of prevalent left heart disease, pulmonary dysfunction, and VTE with abnormalities of PASP, PVR, and PAC were assessed using multivariable logistic regression with the pulmonary vascular measure as the outcome variable and adjusting for age, sex, race, and each of the potential etiologies. Pulmonary dysfunction was further specified as COPD or a restrictive ventilatory abnormality, as pathophysiology of pulmonary vascular disease may differ between these two categories of lung disease [32]. We determined the population attributable risk (PAR) for abnormal pulmonary vascular measures (PASP, PVR, PAC) associated with each potential etiology (left heart disease, pulmonary dysfunction, VTE). We used the prevalence among cases and the odds ratio estimate to calculate the percentage PAR using the following formula [33]: PAR % = $pd_i$ *[$RR_i$-1/$RR_i$], where $pd_i$ is the proportion of total cases in the population arising from the $i$th exposure category and $RR_i$ is the adjusted risk ratio for the $i$th exposure category.

The continuous association of PASP, PVR, and PAC with visit 5 log-transformed NT-proBNP levels was assessed using restricted cubic splines. The number of knots was selected based on minimization of the model AIC (3–6 knots assessed). If the $p$-value for nonlinearity was >0.05 for spline models employing the optimal number of knots, then associations were modeled linearly with 2 knots.

Univariate and multivariable Cox proportional hazards models were employed to assess the association of PASP, PVR, and PAC with incident HF overall, incident HFrEF, incident HFpEF, and the composite of each of these with death post visit 5. Pulmonary vascular measures were modeled as dichotomous (normal, abnormal) and continuous variables. For incident HFpEF, participants developing HF with LVEF <50% or with unknown EF were censored at the time of their HF event, and vice versa for incident HFrEF.

To account for potential bias due to selective attrition related to either unmeasurable TR jet velocity on visit 5 echocardiography or visit 5 non-attendance, we performed a sensitivity analysis using inverse probability of attrition weighting [34,35] (S1 Text). All analyses were performed with STATA 12.0 (College Station, TX). A two-sided $p$-value of <0.05 was considered statistically significant. This study is reported as per the Strengthening the Reporting of Observational Studies in Epidemiology (STROBE) guideline (S1 Checklist).

## Results

The mean age of the 2,810 participants was 76 ± 5 years, 66% were female, and 21% were black (Table 1). Differences between HF-free participants at visit 5 ($n$ = 2,810) and those who were excluded ($n$ = 12,982) are shown in S1 Table. The mean values of PASP, PVR, and PAC were

**Table 1. Clinical characteristics and echocardiographic features of the study cohort.**

| Demographics | Total *n* | Studied cohort |
|---|---|---|
| Age, years | | 76.2 ± 5.2 |
| Male sex, % | 2,810 | 969 (34%) |
| Black, % | 2,810 | 600 (21%) |
| **Field center** | | |
| Forsyth County, NC | 2,810 | 806 (29%) |
| Jackson, MS | | 539 (19%) |
| Minneapolis, MN | | 643(23%) |
| Washington County, MD | | 822 (29%) |
| **Comorbidities** | | |
| Hypertension, % | 2,810 | 2,255 (80%) |
| Diabetes, % | 2,810 | 892 (32%) |
| Obesity, % | 2,795 | 815 (29%) |
| Metabolic Syndrome, % | 2,739 | 1,510 (55%) |
| CKD, % | 2,789 | 744 (27%) |
| Past history of smoking, % | 2,810 | 1,618 (58%) |
| Current smoker, % | 2,706 | 150 (6%) |
| **Prevalent cardiovascular disease** | | |
| CAD, % | 2,810 | 300 (11%) |
| Previous MI, % | 2,647 | 96 (4%) |
| PAD, % | 2,810 | 378 (13%) |
| Previous stroke, % | 2,806 | 73 (3%) |
| Previous Afib, % | 2,810 | 155 (6%) |
| **Physical exam** | | |
| BMI, kg/m$^2$ | 2,795 | 27.8 ± 5.3 |
| SBP, mm Hg | 2,803 | 130 ± 18 |
| DBP, mm Hg | 2,803 | 67 ± 10 |
| HR, bpm | 2,741 | 62 ± 10 |
| **Laboratory values** | | |
| eGFR, mL/min/1.73 m$^2$ | 2,789 | 70.4 ± 16.7 |
| hs-TnT, ng/mL | 2,787 | 0.010 [0.007–0.015] |
| hs-CRP, mg/L | 2,784 | 1.9 [0.9–4.0] |
| **Echocardiographic data** | | |
| Pulmonary vascular hemodynamics | | |
| PASP, mm Hg | 2,810 | 28 ± 5 |
| PVR, WU | 2,798 | 1.7 ± 0.4 |
| PAC, mL/mm Hg | 2,149 | 3.4 ± 1.0 |
| LV structure | | |
| LVEDD, cm | 2,807 | 4.33 ± 0.47 |
| MWT, cm | 2,807 | 0.97 ± 0.13 |
| LVM, mg | 2,806 | 140 ± 40 |
| LVMi, mg/m$^2$ | 2,800 | 77 ± 18 |
| RWT | 2,757 | 0.4 ± 0.1 |
| LV systolic function | | |
| LVEF, % | 2,810 | 66.1 ± 5.8 |
| LV longitudinal strain, % | 2,740 | −18.3 ± 2.4 |
| LV diastolic function | | |
| TDI E' septal, cm/s | 205 | 5.8 ± 1.5 |

(*Continued*)

**Table 1.** (Continued)

| Demographics | Total *n* | Studied cohort |
| --- | --- | --- |
| E/E' septal | 2,802 | 12.3 ± 4.1 |
| LAVi, mL/m$^2$ | 2,793 | 26.3 ± 9.0 |
| RV function | | |
| RVFAC, % | 2,674 | 53 ± 8 |
| TA S', cm/s | 2,795 | 12.0 ± 2.9 |

Values for continuous variables are mean ± SD, except for nonnormally distributed variables, for which median [interquartile range] is shown.

Abbreviations: Afib, atrial fibrillation; BMI, body mass index; CAD, coronary artery disease; CKD, chronic kidney disease; DBP, diastolic blood pressure; eGFR, estimated glomerular filtration rate; HR, heart rate; hs-CRP, high-sensitivity C-reactive protein; hs-TnT, high-sensitivity troponin T; LAVi, left atrial volume index; LV, left ventricular; LVEDD, LV end-diastolic diameter; LVEF, LV ejection fraction; LVM, LV mass; LVMi, LVM index; MI, myocardial infarction; MWT, mean wall thickness; PAC, pulmonary artery compliance; PAD, peripheral artery disease; PASP, pulmonary artery systolic pressure; PVR, pulmonary vascular resistance; RV, right ventricle; RVFAC, RV fractional area change; RWT, relative wall thickness; SBP, systolic blood pressure; TA S', tricuspid annulus; TDI E', Tissue Doppler Imaging–based early diastolic mitral annular velocity; WU, Wood unit.

28 ± 5 mm Hg, 1.7 ± 0.4 Wood unit (WU), and 3.4 ± 1.0 mL/mm Hg, respectively (S1 Fig). PASP was moderately correlated with PVR (r = 0.31; $p < 0.001$) and with PAC (r = −0.40; $p < 0.001$). Pulmonary arterial pressure and resistance increased, and compliance decreased, with increasing age (change per 5 years in late life: PASP: 0.7 mm Hg [95% CI 0.46–0.85], $p < 0.001$; PVR: 0.009 WU [95% CI 0.006–0.012, $p < 0.001$]; PAC: −0.05 mL/mm Hg [95% CI −0.09 to −0.01, $p = 0.013$]). Pulmonary arterial measures were also directly correlated with their corresponding systemic arterial measures independent of demographics, LVEF, LV mass index (LVMi), LAVi, and septal E/E' (associations between: PASP and systolic blood pressure: r = 0.14, $p < 0.001$; mean PAP and mean arterial pressure: r = 0.11, $p = 0.001$; PVR and systemic vascular resistance, r = 0.12, $p < 0.001$; PAC and systemic arterial compliance, r = 0.5 $p < 0.001$; S2 Fig).

From the low-risk reference subgroup (n = 253, mean ± SD age 75 ± 5 years, 71% female, 8% black; S2 Table), the upper reference limits for PASP and PVR were 32 mm Hg and 2.19 WU, respectively, and the lower reference limit for PAC was 2.5 mL/mm Hg (S3 Table). No significant sex-based differences were noted in these reference limits. Applying these reference limits to the overall study population (n = 2,810), 18%, 12%, and 14% of the studied population had an abnormal PASP, PVR, and PAC, respectively (S1 Fig). Abnormal PASP, PVR, and PAC did not demonstrate complete overlap (Fig 1A), with many participants demonstrating abnormalities of only one or two of these measures. At least one pulmonary vascular measure was abnormal in 35% of the study sample.

## Associations of left heart disease, pulmonary dysfunction, and VTE with pulmonary vasculature dysfunction

Left heart dysfunction was present in 29% of the study population, pulmonary dysfunction was present in 45% (COPD in 28%; restrictive ventilatory defect in 12%; mixed in 5%), and prior VTE history was present in 3% (Fig 1B; Fig 2; S4 Table). Left heart dysfunction demonstrated the greatest magnitude of association with abnormally high PASP and was also associated with a higher prevalence of abnormally low PAC (Fig 2; Table 2). Similar findings were noted when using commonly employed cutoff values of 40 mm Hg for PASP and 3.0 WU for

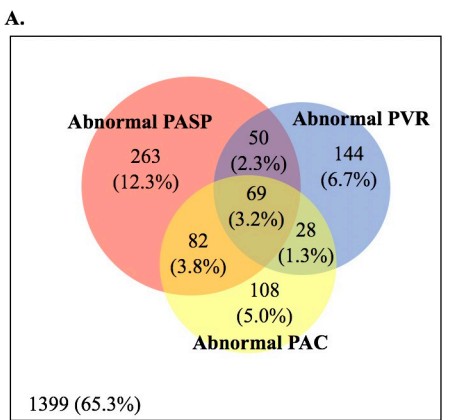
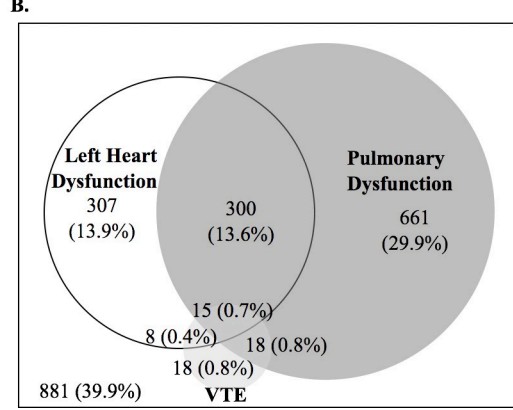

**Fig 1. Prevalence of abnormal pulmonary vasculature measures and pulmonary functional.** (A) Overlap between abnormal PASP, PVR, and PAC defined using ARIC-based reference limits. (B) Overlap between left heart disease, pulmonary dysfunction, and VTE. ARIC, Atherosclerosis Risk in Communities; PAC, pulmonary arterial compliance; PASP, pulmonary artery systolic pressure; PVR, pulmonary vascular resistance; VTE, venous thromboembolism.

PVR (S5 Table) and when further excluding an additional 227 participants with moderate or severe dyspnea and European Society of Cardiology Heart Failure Association (ESC HFA) criteria for HFpEF (*n* = 196) and also those with LVEF <50% (*n* = 31; S6 Table).

Pulmonary dysfunction demonstrated associations of similar magnitude with all pulmonary vascular measures, which appeared to be driven by associations with restrictive ventilatory deficit as opposed to COPD (S4 Table). Prior VTE was not associated with abnormalities in any pulmonary vascular measure. Left heart dysfunction demonstrated the highest PAR for abnormally high PASP (20%), whereas pulmonary dysfunction had the highest PAR for abnormalities of PAC (19%; Fig 2; Table 2).

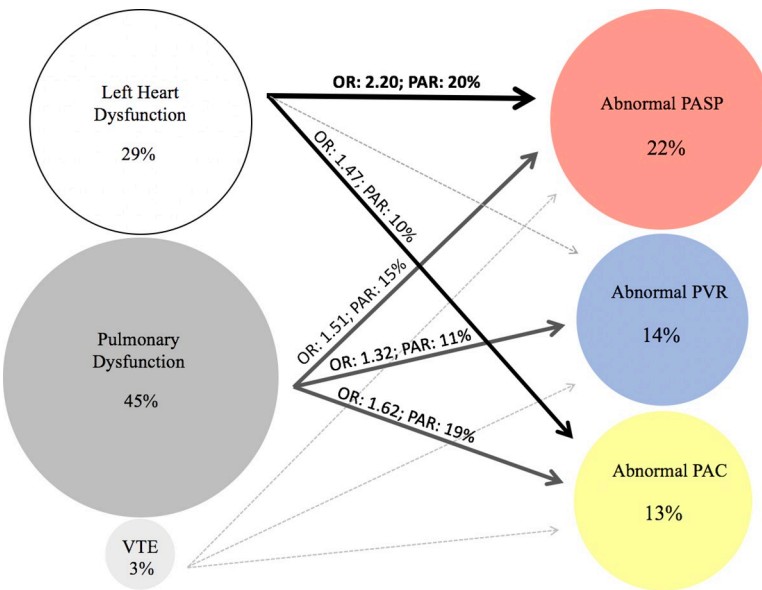

**Fig 2. Prevalence, association, and PAR of left heart disease, pulmonary dysfunction, and VTE for abnormalities of PASP, PVR, and PAC.** OR, odds ratio; PAC, pulmonary arterial compliance; PAR, population attributable risk; PASP, pulmonary artery systolic pressure; PVR, pulmonary vascular resistance; VTE, venous thromboembolism.

**Table 2. Prevalence of LHD, pulmonary dysfunction, and VTE and their association with abnormal PASP, PVR, or PAC.**

| Abnormal measures | N abnormal | OR (95% CI) | p-Value | PAR (95%) |
|---|---|---|---|---|
| **Abnormal PASP** | Total n = 2,810 | | | |
| LHD | 798 (28%) | 2.20 (1.75–2.76) | <0.001 | 19.9 (13.7–25.6) |
| Pulmonary dysfunction | 994 (35%) | 1.51 (1.21–1.89) | <0.001 | 14.9 (6.5–22.6) |
| Prior VTE | 76 (3%) | 1.32 (1.72–2.44) | 0.37 | 6.8 (−0.9 to 2.3) [NS] |
| **Abnormal PVR** | Total n = 2,798 | | | |
| LHD | 793 (28%) | 1.06 (0.80–1.42) | 0.68 | 1.6 (−6.5 to 9.1) [NS] |
| Pulmonary dysfunction | 988 (35%) | 1.32 (1.01–1.72) | 0.046 | 10.7 (−0.4 to 20.5) [NS] |
| Prior VTE | 76 (3%) | 1.16 (0.53–2.52) | 0.71 | 3.6 (−1.6 to 2.3) [NS] |
| **Abnormal PAC** | Total n = 2,149 | | | |
| LHD | 633 (29%) | 1.47 (1.09–1.97) | 0.011 | 10.2 (1.6–18.0) |
| Pulmonary dysfunction | 766 (36%) | 1.62 (1.21–2.16) | 0.001 | 19.0 (7.0–29.5) |
| Prior VTE | 62 (3%) | 0.45 (0.16–1.29) | 0.14 | −1.7 (−3.3 to −0.04) [NS] |

Logistic regression models are used to estimate the OR and p-values of each pulmonary vasculature dysfunction for having abnormal pulmonary measures. The model contains age, sex, race, visit center LHD, pulmonary dysfunction, and prior VTE.

Abbreviations: LHD, left heart dysfunction; NS, not significant; OR, odds ratio; PAC, pulmonary arterial compliance; PAR, population attributable risk; PASP, pulmonary artery systolic pressure; PVR, pulmonary vascular resistance; VTE, venous thromboembolism.

## Relationship of pulmonary vascular hemodynamics with NT-proBNP, RV function, and incident HF

Higher PASP, higher PVR, and lower PAC were each associated with higher circulating concentrations of NT-proBNP (Fig 3A). These associations persisted after adjustment for

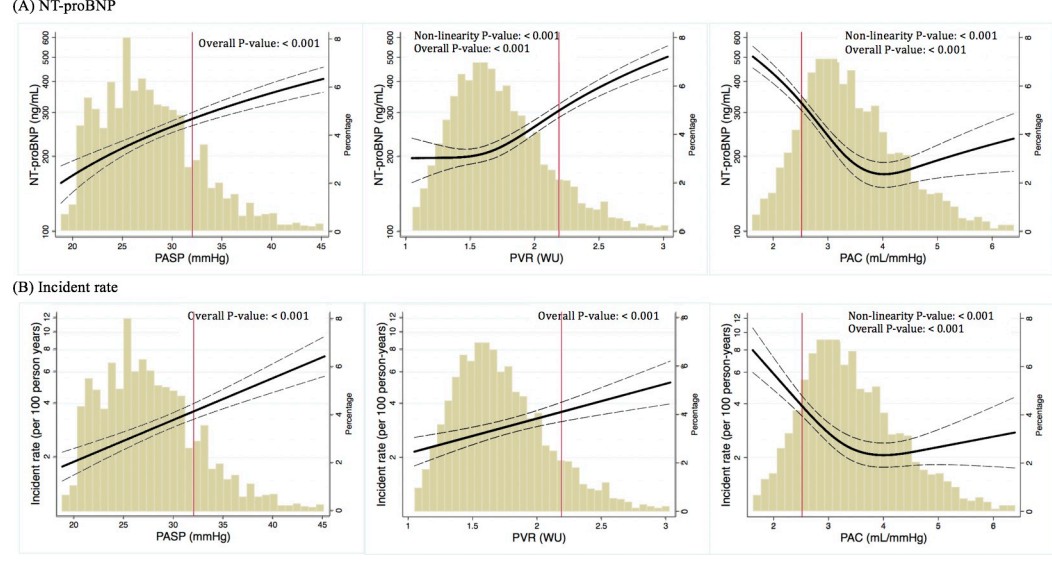

**Fig 3.** Association of pulmonary hemodynamic measures with (A) plasma NT-proBNP concentrations and (B) incident HF or death post visit 5. Histogram shows the distribution of the primary hemodynamic variable in the study sample. Black dotted line represents 95% confidence interval. Red line indicates the ARIC-based reference value for abnormal for that pulmonary vasculature measure (see text for details). p-Values are derived from cubic spline model adjusted for age, sex, race, visit center, and LV measures (LVEF, LAVi, LVMi, and septal E/E'). ARIC, Atherosclerosis Risk in Communities; HF, heart failure; LAVi, left atrial volume index; LV, left ventricular; LVEF, LV ejection fraction; LVMi, LV mass index; NT-proBNP, N-terminal prohormone brain natriuretic peptide; PAC, pulmonary arterial compliance; PASP, pulmonary artery systolic pressure; PVR, pulmonary vascular resistance; WU, Wood unit.

participant demographics (age, sex, race, visit center, and BMI), hypertension, diabetes, and LV measures including LVEF, LVMi, LAVi, and E/e' ($p < 0.001$ for PASP, $p < 0.001$ for PVR, and $p < 0.001$ for PAC; Fig 3A). Higher PASP and lower PAC were associated with lower TA s' (overall $p < 0.001$ and 0.02, respectively; S3B Fig), whereas higher PVR and lower PAC were both associated with lower RVFAC ($p < 0.001$ for PVR and $p < 0.001$ for PAC; S3A Fig).

Total of 446 cases of death or HF admission occurred over a mean ± SD follow-up of 5.3 ± 1.2 years (range of 6.5 years), 161 participants developed incident HF (71 HFpEF, 65 HFrEF, 25 HF with unknown LVEF), and 344 died. Incident HF or death occurred in 446 (59 with incident HF prior to death: 24 HFpEF, 26 HFrEF, 9 HF with unknown LVEF). In models adjusted for age, sex, race, visit center, BMI, hypertension, diabetes, and LV measures (LVEF, LAVi, LVMi, and septal E/E'), higher PASP, higher PVR, and lower PAC were each predictive of an increased risk of death or incident HF (Fig 3B, S7 Table [model 3]). After additional adjustment for PASP, abnormal PVR and PAC were no longer significantly associated with incident HF or death (S7 Table [model 4]). Abnormal PASP was predictive of incident HFpEF but not HFrEF. In contrast, abnormal PAC appeared to demonstrate modestly stronger association with incident HFrEF than HFpEF whereas abnormal PVR did not predict either (Fig 4). Directionally similar results were noted for the composite with death, suggesting that results were not significantly impacted by the competing risk of death. Similar findings were also observed when modeling the pulmonary vascular measures as continuous variables (S8 Table), when using inverse probability weights (S9 Table), when further excluding an additional 227 participants with moderate or severe dyspnea and ESC HFA criteria for HFpEF and those with LVEF <50% (S10 Table), and when using cutoff values of 40 mm Hg for PASP and 3.0 WU for PVR, although power was limited for associations with incident HFpEF and HFrEF (S11 Table).

A greater number of abnormal pulmonary vascular measures was associated with progressively higher circulating NT-proBNP levels (S4A Fig) and greater incidence of HF (adjusted

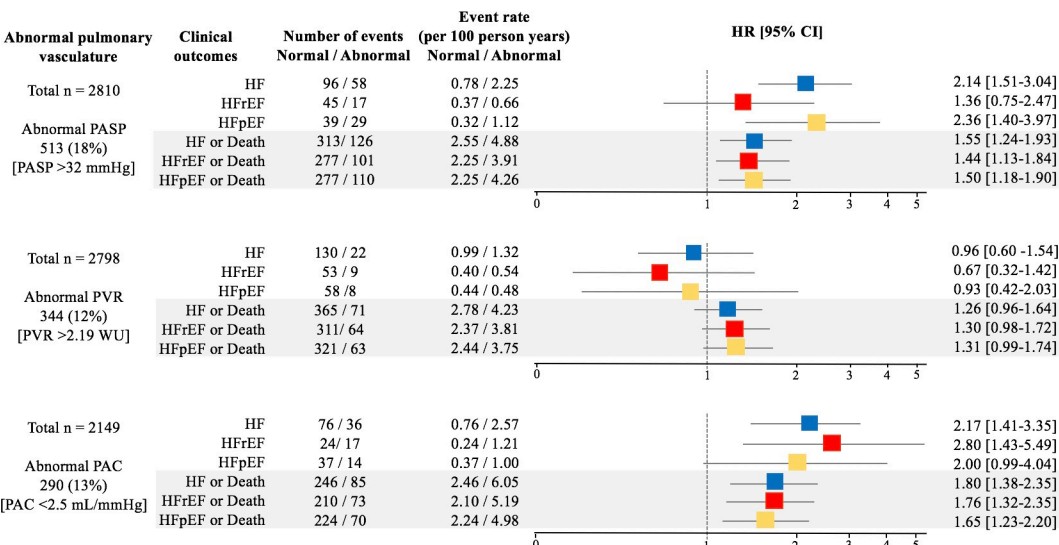

**Fig 4. Association of abnormalities of pulmonary vascular measures (PASP, PVR, PAC) with incident HF overall, incident HFpEF, incident HFrEF, or the composite of death with each of these.** Models are adjusted for age, sex, race, visit center, BMI, hypertension, diabetes, LVEF, LAVi, LVMi, and septal E/e'. BMI, body mass index; HF, heart failure; HFpEF, HF with preserved LVEF; HFrEF, HF with reduced LVEF; HR, hazard ratio; LAVi, left atrial volume index; LV, left ventricular; LVEF, LV ejection fraction; LVMi, LV mass index; PAC, pulmonary arterial compliance; PASP, pulmonary artery systolic pressure; PVR, pulmonary vascular resistance; WU, Wood unit.

hazard ratio relative to no abnormalities: 1.8 [95% CI 1.2–2.8, $p$ = 0.007] for one abnormality; 1.8 [1.0–3.3, $p$ = 0.07] for two abnormalities; 4.0 [1.9–8.3, $p$ < 0.001] for three abnormalities; $p$ for trend < 0.001 [95% CI 1.2–1.9]; S4B Fig).

## Discussion

Pulmonary vascular function is known to change with age, and pulmonary vascular dysfunction is emerging as an important contributor to HF in general and HFpEF in particular. This study defines the prevalence of pulmonary vascular dysfunction—comprehensively characterized based on pressure, resistance, and compliance—in an elderly community-based sample, quantifies its association with cardiac versus pulmonary dysfunction, and establishes its prognostic importance for incident HFpEF and HFrEF. Worse PASP, PVR, and PAC were each associated with higher NT-proBNP and greater risk of incident HF independent of LV measures. Our data therefore build on previous reports showing functional [36] and histological changes of pulmonary vessels in patients with HF [4,37], by extending this concept to the elderly at risk for incident HF, using a comprehensive characterization of pulmonary vascular function. PH, the most recognized clinical manifestation of pulmonary vascular dysfunction, was most strongly related to left heart dysfunction and appeared to be particularly predictive of incident HFpEF. Abnormalities of PVR and PAC were most closely related to pulmonary dysfunction. Abnormal PAC demonstrated similar associations with incident HFrEF and HFpEF but appeared modestly more strongly predictive of HFrEF. These findings suggest that impairments in pulmonary vascular dysfunction precede clinical HF, and abnormalities of different measures of pulmonary vasculature may differentially predict incident HF phenotype.

Similar to prior studies, PASP increased with age [7,38,39,40], as did PVR, whereas PAC declined, paralleling age-related changes observed in the systemic arteries [7]. Indeed, in our study, pulmonary and systemic arterial hemodynamics were significantly correlated, independent of measures of LV structure and function, arguing for coupled age-related pulmonary and systemic vascular dysfunction. The large relative contribution of left heart dysfunction to elevated PASP in this study is consistent with existing data suggesting that left-sided heart dysfunction is the most prevalent cause of PH worldwide [41,42]. The association between left heart dysfunction and low PAC is also consistent with the known relationship between increased left-sided filling pressures and worse PAC for any given PVR [43,44]. In the pulmonary arterial circulation, the majority of compliance originates in the distal vasculature [45,46], compliance and resistance have an inverse hyperbolic relationship with a constant resistance-compliance product (RC time) [22,43,45,47], and higher left atrial pressure exaggerates reductions in compliance for any increase in resistance [43].

The generally mild degree of COPD in our sample likely accounts for the modest association of COPD with pulmonary vascular measures, as the prevalence of PH in mild COPD is low [32]. The prevalence of a restrictive ventilatory deficit was comparable to other elderly cohorts [48] and was robustly related to all pulmonary vascular measures, independent of LV measures. These findings suggest that a restrictive ventilatory deficit does not simply reflect changes associated with left heart disease [49] and support the need for studies evaluating the potential impact of interstitial pulmonary parenchymal disease and pulmonary respiratory muscle weakness on pulmonary hemodynamics in the elderly. The lack of association between VTE and abnormal pulmonary hemodynamics was expected, as chronic thromboembolic PH is only a rare complication of pulmonary embolism [50].

Despite intense interest, data regarding the prognostic importance of pulmonary hemodynamics in the community are limited and largely restricted to PASP. Among 1,413 participants in Olmsted county (mean age of 63 ± 11 years), each 10–mm Hg increase in PASP above 23

mm Hg was associated with a 2.7-fold increase in mortality [7]. Among 3,125 African American participants in the Jackson Heart Study (mean age of 56 ± 13 years), a PASP >33 mm Hg was associated with a greater risk of incident HF [51]. Our study now extends the independent prognostic importance of higher PASP for incident HF or death to an elderly biracial cohort (mean ± SD age of 76 ± 2 years; 21% black). We further demonstrate that elevated pulmonary pressure in late life appears more strongly associated with incident HFpEF than HFrEF. While the relationship between PH and HFpEF in patients with prevalent HFpEF is potentially bidirectional, the differential association of elevated pulmonary pressure with HFpEF in our study—independent of LV structure and function—suggests an important pathophysiologic role for PH in development of HFpEF.

Beyond PASP, our study—for the first time, to our knowledge—describes the prognostic value of PVR and PAC in an elderly community-based cohort. These are two inversely related measures of pulmonary hydraulic load [3]. PVR, a measure of nonpulsatile load, is commonly used in clinical practice to assess pulmonary vascular disease and is associated with prognosis in HF [52]. More recently, PAC has gained attention as a measure of arterial distensibility and pulsatile load that changes earlier in the development of pulmonary vascular disease than PVR. Lower PAC in patients with HF has been linked to worse prognosis [5,53]. The value of assessing these complementary hemodynamic measures is illustrated by our observation that abnormalities of PASP, PVR, and PAC did not completely overlap, that these measures may differentially predicted incident HFpEF and HFrEF, and that a greater number of abnormal measures was associated with heightened risk of both incident HF. Our data suggest that the comprehensive evaluation of pulmonary hemodynamics beyond PASP provides more detailed and prognostically relevant pulmonary vascular phenotyping. This is particularly important, as the results of prior analyses using hierarchical clustering approaches suggest that pulmonary vascular disease identifies a distinct HFpEF phenogroup [4,54,55]. Furthermore, at least eight ongoing phase 2–3 randomized controlled trials are testing agents targeting PH in HFpEF [56]. Comprehensive assessment of pulmonary arterial pressure, resistance, and compliance may ultimately help clinicians better identify older adults at heighted risk for HF and better target patients for interventions to prevent and treat HF through more detailed phenotyping of the pulmonary vasculature.

PASP, PVR, and PAC were each also associated with important intermediary HF measures, including NT-proBNP and RV function. Worse values of each measure were associated with higher NT-proBNP, an established biomarker of HF risk. Both high PVR and lower PAC were associated with worse RV systolic function, whereas PASP—within the range represented in this study sample—was not associated with RVFAC.

Our study had several limitations. TR jet measurement of adequate quality was feasible in only a subset (57%) of participants, but this rate is similar to that reported in other studies [7,51]. Furthermore, sensitivity analyses incorporating inverse probability of attrition weights to account for participants without measurable TR jet, or alive but not attending visit 5, demonstrated similar results to our primary analysis (S8 Table), suggesting generalizability of our findings. We used echocardiography instead of gold-standard invasive right heart catheterization to assess pulmonary hemodynamics, as broad implementation of invasive right heart catheterization in a large HF-free elderly cohort would be neither feasible nor ethical. The echocardiographic formulas used in our study to estimate the pulmonary hemodynamic variables were previously validated against right heart catheterization but remain imperfect estimations [11,18]. In particular, the agreement between echocardiographic and invasive hemodynamics measures is poor in patients with moderate to severe PH [57], but this was uncommon in the ARIC sample. Nonetheless, the use of echocardiography-based measures could result in appreciable misclassification. PVR and PAC are highly dependent on TR

velocity, and the limited number participants with measurable values for PAC in particular may have led to overfitting of some models. We used spirometric data to classify the lung disease instead of plethysmography, the gold standard to accurately measure lung volumes [58]. In addition to feasibility issues of applying plethysmography in a large cohort, FVC is a commonly used proxy for total lung capacity in clinical practice. The majority of participants in this study were women, which may limit the generalizability of these findings. The limited number of participants in some subgroups and the relatively short follow-up limited statistical power for analyses related to incident death or HF. Nevertheless, this study is one of the largest and most comprehensive community-based analyses of pulmonary hemodynamics and their prognostic relevance. Furthermore, additional analysis of the association of pulmonary hemodynamic measures with NT-proBNP concentration, a robust and validated surrogate biomarker of HF risk, demonstrated concordant results.

In a large community-based cohort of older adults free of HF, subclinical left heart dysfunction and pulmonary dysfunction were both significantly related to worse PASP, PVR, and PAC. Worse PASP and PAC were each associated with greater risk of incident HF independent of LV measures. Elevated pulmonary pressure was associated with incident HFpEF whereas abnormal PAC appeared more predictive of incident HFrEF. A greater number of abnormal measures was associated with greater risk of incident HF. These findings suggest that impairments in pulmonary vascular function precede clinical HF and that comprehensive pulmonary hemodynamic evaluation may identify pulmonary vascular phenotypes that differentially predict HF phenotypes.

## Supporting information

**S1 Checklist. STROBE checklist.**
(DOCX)

**S1 Fig. Distribution of pulmonary hemodynamics in the study sample.** Density and cumulative distributions are presented. Dashed lines represent the reference limit derived based on the 10th percentile of the low-risk subset of participants for PASP and PVR. For PAC, the 90th percentile of the low-risk subset of participants were used as reference limit. PAC, pulmonary arterial compliance; PASP, pulmonary artery systolic pressure; PVR, pulmonary vascular resistance.
(TIF)

**S2 Fig. Scatterplot depicting the associations between pulmonary and systemic hemodynamics.** (A) PASP and SBP; (B) MPAP and MAP; (C) PVR and SVR; and (D) PAC and SAC. *p*-Values were derived from multivariable regression model adjusted for age, sex, race, visit center, BMI, hypertension, diabetes, and LV measures (LVEF, LAVi, LVMi, and septal E/E'). BMI, body mass index; LAVi, left atrial volume index; LV, left ventricular; LVEF, LV ejection fraction; LVMi, LV mass index; MAP, mean arterial pressure; MPAP, mean pulmonary arterial pressure; PAC, pulmonary arterial compliance; PASP, pulmonary artery systolic pressure; PVR, pulmonary vascular resistance; SAC, systemic arterial compliance; SVR, systemic vascular resistance.
(TIF)

**S3 Fig. Relationship between pulmonary hemodynamic measures (PASP, PVR, PAC) and RV function (RVFAC, TA s').** *p*-Values were derived from cubic spline regression model adjusted for age, sex, race, visit center, BMI, hypertension, diabetes, LVEF, LAVi, LVMi, and septal E/e'. BMI, body mass index; LAVi, left atrial volume index; LVEF, left ventricular ejection fraction; LVMi, left ventricular mass index; PAC, pulmonary arterial compliance; PASP,

pulmonary artery systolic pressure; PVR, pulmonary vascular resistance; RV, right ventricular; RVFAC, RV fractional area change; TA s', tricuspid annulus.
(TIF)

**S4 Fig. Relationship between number of abnormal pulmonary vascular measures and (A) NT-proBNP plasma concentrations, (B) incidence of HF or the composite of HF or death.** Event rate in per 100 person-years and error bars presenting the upper limit of 95% confidence interval. HF, heart failure; NT-proBNP, N-terminal prohormone brain natriuretic peptide.
(TIF)

**S1 Table. Clinical and echocardiographic characteristics of participants included in the study versus those not included in the study.** All continuous variables are described as mean ± SD. Nonparametric values are presented with median and interquartile range in square brackets. *p*-Values are derived from ANOVA for continuous variables, Pearson chi-squared test for binary and categorical variables, and Kruskal-Wallis test for nonparametric continuous variables. Afib, atrial fibrillation; BMI, body mass index; CAD, coronary artery disease; CKD, chronic kidney disease; DBP, diastolic blood pressure; eGFR, estimated glomerular filtration rate; HR, heart rate; hs-CRP, high-sensitivity C-reactive protein; hs-TnT, high-sensitivity troponin T; LAVi, left atrial volume index; LVEDD, left ventricular end-diastolic diameter; LVEF, left ventricular ejection fraction; LVM, left ventricular mass; LVMi, LVM index; MI, myocardial infarction; MWT, mean wall thickness; PAD, peripheral artery disease; RVFAC, right ventricle fractional area change; RWT, relative wall thickness; SBP, systolic blood pressure; TA S', tricuspid annulus.
(DOCX)

**S2 Table. Clinical and echocardiographic characteristics of study participants in low-risk subgroup compared to those not in the low-risk subgroup.** All continuous variables are described as mean ± SD. Nonparametric values are presented with median and interquartile range in square brackets. *p*-Values are derived from ANOVA for continuous variables, Pearson chi-squared test for binary and categorical variables, and Kruskal-Wallis test for nonparametric continuous variables. Afib, atrial fibrillation; BMI, body mass index; CAD, coronary artery disease; CKD, chronic kidney disease; DBP, diastolic blood pressure; eGFR, estimated glomerular filtration rate; HR, heart rate; hs-CRP, high-sensitivity C-reactive protein; hs-TnT, high-sensitivity troponin T; MI, myocardial infarction; PAD, peripheral artery disease; SBP, systolic blood pressure.
(DOCX)

**S3 Table. Percentile limits for pulmonary hemodynamic measures among the low-risk reference subgroup (*n* = 253).** The 10th, 50th, and 90th percentile values with associated 95% confidence intervals derived from quantile regression models in the low-risk reference subgroup overall and separately by sex. PAC, pulmonary arterial compliance; PAP, pulmonary arterial pressure; PASP, pulmonary arterial systolic pressure; PVR, pulmonary vascular resistance; TR, tricuspid regurgitation.
(DOCX)

**S4 Table. Prevalence of LHD, pulmonary dysfunction (COPD and restrictive lung disease), and VTE and their association with measures of pulmonary vascular dysfunction.** Estimates for COPD and restrictive lung disease as measures of pulmonary dysfunction are provided separately. Logistic regression models are used to estimate the odds ratio and *p*-values of each pulmonary vasculature dysfunction for having abnormal pulmonary measures. The model contains age, sex, race, visit center, LHD, COPD, restrictive lung disease, and prior

VTE. COPD, chronic obstructive pulmonary disease; LHD, left heart dysfunction; PAR, populational attributable risk; VTE, venous thromboembolism.
(DOCX)

**S5 Table. Prevalence of LHD, pulmonary dysfunction, and VTE and their association with measures pulmonary vascular dysfunction using common clinical reference limits to define abnormal PASP (>40 mm Hg) and PVR (>3.0 WU).** Odds values and *p*-values are derived from multivariable logistic models containing age, sex, race, visit center, LHD, pulmonary dysfunction, and prior VTE. LHD, left heart disease; NS, not significant; PAR, population attributable risk; PASP, pulmonary artery systolic pressure; PVR, pulmonary vascular resistance; VTE, venous thromboembolism; WU, Wood unit.
(DOCX)

**S6 Table. Prevalence of LHD, pulmonary dysfunction, and VTE and their association with abnormal PASP, PVR, or PAC after further excluding an additional 227 participants with moderate or severe dyspnea and ESC HFA criterial for HFpEF and those with LVEF <50%.** Odds values and *p*-values are derived from multivariable logistic models containing age, sex, race, visit center, LHD, pulmonary dysfunction, and prior VTE. ESC HFA, European Society of Cardiology Heart Failure Association; HFpEF, incident HF with preserved LVEF; LHD, left heart disease; LVEF, left ventricular ejection fraction; NS, not significant; PAC, pulmonary arterial compliance; PAR, population attributable risk; PASP, pulmonary artery systolic pressure; PVR, pulmonary vascular resistance; VTE, venous thromboembolism.
(DOCX)

**S7 Table. Association of pulmonary hemodynamic measures with incident HF or death post visit 5.** *p*-Values were derived from multivariable Cox regression analysis. Model 1 adjusts for age, sex, race, and visit center. Model 2 adjusts for LVEF, LAVi, LVMi, and septal E/e' in addition to model 1. Model 3 adjusts for hypertension, diabetes, and body mass index in addition to model 2. Model 4 adjusts for PASP in addition to Model 3. HF, heart failure; LAVi, left atrial volume index; LVEF, left ventricular ejection fraction; LVMi, left ventricular mass index; PASP, pulmonary artery systolic pressure.
(DOCX)

**S8 Table. Continuous association of pulmonary hemodynamic measures with incident HFrEF, HFpEF, and the composite of each of these with death (per 1 SD change).** *p*-Values were derived from multivariable Cox regression models. Model 1 adjusts for age, sex, race, and visit center. Model 2 adjusts for LVEF, LAVi, LVMi, and septal E/e' in addition to model 1. Model 3 adjusts for hypertension, diabetes, and body mass index in addition to model 2. HFrEF, heart failure with reduced ejection fraction (LVEF < 50%); HFpEF, heart failure with preserved ejection fraction (LVEF ≥ 50%); LAVi, left atrial volume index; LVEF, left ventricular ejection fraction; LVMi, left ventricular mass index.
(DOCX)

**S9 Table. Association of pulmonary hemodynamic measures with incident HF or death post visit 5 using inverse probability weights.** *p*-Values were derived from inverse probability weighted multivariable Cox regression analysis. Model 1 adjusts for age, sex, race, and visit center. Model 2 adjusts for LVEF, LAVi, LVMi, and septal E/e' in addition to model 1. Model 3 adjusts for hypertension, diabetes, and body mass index in addition to model 2. HF, heart failure; LAVi, left atrial volume index; LVEF, left ventricular ejection fraction; LVMi, left ventricular mass index.
(DOCX)

**S10 Table. Association of abnormalities of pulmonary vascular measures (PASP, PVR, PAC) with incident HF overall, incident HFpEF, incident HFrEF, or the composite of death with each of these when further excluding an additional 227 participants with moderate or severe dyspnea and ESC HFA criterial for HFpEF and those with LVEF <50%.** *p*-Values were derived from multivariable Cox regression model adjusted for age, sex, race, visit center, BMI, hypertension, diabetes, LVEF, LAVi, LVMi, and septal E/e'. BMI, body mass index; ESC HFA, European Society of Cardiology Heart Failure Association; HF, heart failure; HFrEF, heart failure with reduced ejection fraction (LVEF < 50%); HFpEF, heart failure with preserved ejection fraction (LVEF ≥ 50%); LAVi, left atrial volume index; LVEF, left ventricular ejection fraction; LVMi, left ventricular mass index; PAC, pulmonary arterial compliance; PASP, pulmonary artery systolic pressure; PVR, pulmonary vascular resistance. (DOCX)

**S11 Table. Association of pulmonary hemodynamic measures based on established reference limits with incident HF or death post visit 5.** *p*-Values were derived from multivariable Cox regression model adjusted for age, sex, race, visit center, BMI, hypertension, diabetes, LVEF, LAVi, LVMi, and septal E/e'. BMI, body mass index; HF, heart failure; HFrEF, heart failure with reduced ejection fraction (LVEF < 50%); HFpEF, heart failure with preserved ejection fraction (LVEF ≥ 50%); LAVi, left atrial volume index; LVEF, left ventricular ejection fraction; LVMi, left ventricular mass index. (DOCX)

**S1 Text.**
(DOCX)

## Acknowledgments

The authors thank the staff and participants of the ARIC study for their important contribution.

## Author Contributions

**Conceptualization:** Kanako Teramoto, Mário Santos, Amil M. Shah.

**Data curation:** Jenine E. John.

**Formal analysis:** Kanako Teramoto, Mário Santos, Brian Claggett.

**Funding acquisition:** Amil M. Shah.

**Methodology:** Kanako Teramoto, Mário Santos.

**Supervision:** Amil M. Shah.

**Visualization:** Kanako Teramoto.

**Writing – original draft:** Kanako Teramoto, Mário Santos, Amil M. Shah.

**Writing – review & editing:** Kanako Teramoto, Mário Santos, Brian Claggett, Jenine E. John, Scott D. Solomon, Dalane Kitzman, Aaron R. Folsom, Mary Cushman, Kunihiro Matsushita, Hicham Skali, Amil M. Shah.

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
