## [Decision Letter · Decision Letter 0]

5 Feb 2020

Dear Dr. Shah,

Thank you very much for submitting your manuscript "Prevalence, Contributors, and Consequences of Pulmonary Vascular Dysfunction Among Elderly Persons in the Community" (PMEDICINE-D-19-04388) for consideration at PLOS Medicine. 

Your paper was evaluated by a senior editor and discussed among the editors here. It was also discussed with an academic editor with relevant expertise, and sent to independent reviewers, including a statistical reviewer. The reviews are appended at the bottom of this email and any accompanying reviewer attachments can be seen via the link below:

[LINK]

In light of these reviews, I am afraid that we will not be able to accept the manuscript for publication in the journal in its current form, but we would like to consider a revised version that addresses the reviewers' and editors' comments. Obviously we cannot make any decision about publication until we have seen the revised manuscript and your response, and we plan to seek re-review by one or more of the reviewers. 

We expect to receive your revised manuscript by Feb 19 2020 11:59PM. Please email us (plosmedicine@plos.org) if you have any questions or concerns.

We look forward to receiving your revised manuscript. 

Sincerely,

Louise Gaynor-Brook, MBBS PhD

Associate Editor 

PLOS Medicine

plosmedicine.org

General comment: Please fully define abbreviations at first use in the Abstract, and again in the main text of your manuscript 

Please revise your title according to PLOS Medicine's style, placing the study design in the subtitle (ie, after a colon). We suggest “Pulmonary vascular dysfunction among people aged over 65 years in the community (ARIC): a cross-sectional analysis” or similar. NB) Age of over 65 has been taken from line 81 - please correct age based on the lowest age among all participants included in this study

Abstract Background: Please expand upon the context of why the study is important. 

Line 30 - please define LVEF. Please clarify what is meant by (LVEF ≥ or <50%)

Abstract Methods and Findings:

Please include further details of the study setting and main/secondary outcome measures.

Please quantify the main results (with 95% CIs and p values).

Line 34/35 - please omit one instance of the word ‘prospective’ to avoid repetition 

Line 35 - please clarify what communities

Line 37 - please define ‘elderly’ by specifying the age of participants 

Line 38 - please provide dates between which data was collected for this study

Line 40 - please define fully PVR and PAC

Line 41 - please define what 76±5 years represents e.g. mean ± SD

Line 42 - please define WU 

Line 49 - please define HFpEF and HFrEF

In the last sentence of the Abstract Methods and Findings section, please describe the main limitation(s) of the study's methodology.

Please begin your Abstract Conclusions with "In this study, we observed ..." or similar. Please address the study implications, emphasizing what is new without overstating your conclusions.

Line 55/336 - ‘precede’ would be more conventional

At this stage, we ask that you include a short, non-technical Author Summary of your research to make findings accessible to a wide audience that includes both scientists and non-scientists. The Author Summary should immediately follow the Abstract in your revised manuscript. This text is subject to editorial change and should use non-identical language distinct from the scientific abstract. Please see our author guidelines for more information: https://journals.plos.org/plosmedicine/s/revising-your-manuscript#loc-author-summary

Please include a final bullet point under ‘What do these findings mean?’ to describe the main limitations(s) of the study.

Introduction

Please expand upon the need for and potential importance of your study. Indicate whether your study is novel and how you determined that. If there has been a systematic review of the evidence related to your study (or you have conducted one), please refer to and reference that review and indicate whether it supports the need for your study. 

Line 63 / 64 - please define PVR, PAC and LV 

Line 67 - please define HF

Methods

Please ensure that the study is reported according to the STROBE guideline, and include the completed STROBE checklist as Supporting Information. Please add the following statement, or similar, to the Methods: "This study is reported as per the Strengthening the Reporting of Observational Studies in Epidemiology (STROBE) guideline (S1 Checklist)." 

Did your study have a prospective protocol or analysis plan? Please state this (either way) early in the Methods section. If a prospective analysis plan was used in designing the study, please include the relevant prospectively written document with your revised manuscript as a Supporting Information file to be published alongside your study, and cite it in the Methods section. If no such document exists, please make sure that the Methods section transparently describes when analyses were planned, when/why any data-driven changes to analyses took place, and what those changes were. 

Where interviews were conducted, please provide a copy of the proforma(s) used in your supplementary information. 

Line 90 - please provide dates between which data was collected for this study

Results

Please provide 95% CIs in addition to p values.

Line 218 - please provide p values for each of the measures, rather than grouping all as P ≤0.003

Line 251 / 278 / 280 - please provide p values for each of the measures, rather than grouping all as P ≤0.001

Line 277 - Please indicate which factors are adjusted for

Please define the length of follow up (eg, in mean, SD, and range).

Lines 282-284 - please clarify whether the 425 participants who died are included in the 547 participants for whom ‘ Death or HF hospitalization occurred’. Please clarify how all numbers presented here relate to each other

All tables and figures - Please define all abbreviations used in the legend of each table/figure. When a p value is given, please specify the statistical test used to determine it, in the table/figure legend.

All tables - Please present numerators and denominators, in addition to percentages. 

Table 1 - Please ensure that it is clear what results including ± represent (e.g. standard deviation) 

Figure 1 - Please incorporate panel C alongside panels A and B into one figure, as Figure 1; or divide into two separate figures entirely

Figures 3 and S4 - please avoid using red and green together, to improve accessibility to those with colour blindness 

Discussion 

Please discuss the implications of your study and next steps for research, clinical practice, and/or public policy.

Line 391 - please revise sentence beginning ‘This is particularly important...’ to clarify meaning

Please remove subheading for Conclusion 

Comments from the reviewers:

Reviewer #1: Statistical review

This paper reports a prospective cohort study assessing association between echocardiography measurements and cardiovascular outcomes. I had some comments on the statistical methods and reporting, which are provided below.

1. Abstract: what are the figures following the +/- standard deviations, or standard errors, or something else? This is also applicable to the results.

2. Abstract/results - if the abnormality limits were below 10th or above 90th percentiles, then wouldn't 20% of participants be expected to be 'abnormal' even if they were consistent with healthy individuals? This also is relevant to the results on bottom paragraph of page 13 - this actually seems like there is lower abnormality than there should be.

3. Abstract - I would recommend summaries of effect sizes of association, plus confidence intervals, between PASP, PVR and PAC and time to incident HF/death are given. For the first time a confidence interval is given (presumably the square bracket ranges are CIs), I'd mention that it is a confidence interval.

4. Abstract - the acronyms on line 49 are not defined in the abstract (nor in the paper).

5. Line 104 - would be useful additional information to briefly summarise the reproducibility metrics.

6. Line 170 - I see later that the amount of follow-up was not the same for all patients (line 283). If follow-up differs substantially then might a time to event analysis be more suitable than logistic regression (which does not account for differential follow-up)?

7. Line 178/results - Good that confidence intervals were provided for PARs - how were they calculated? How should a 'non-significant' PAR in table 2 be interpreted? I'm unsure marking as NS I informative.

8. Statistical methods: were covariates used in the multivariable models missing in a non-negligible proportion of patients? If so please justify the missing data assumptions made.

9. Line 214-215: I would recommend providing confidence intervals rather than '+/-' values.

10. Table 2 - minor type but I assume PAR for prior VTE abnormal PASP should be (-0.2,3.0)

James Wason

Reviewer #2: In the current work the authors evaluated the contributors to and prognostic importance of pulmonary vascular dysfunction among heart failure-free persons in the community in a cross-sectional and time to event analyses in the ARIC cohort.

Specifically, they assessed the prevalence and correlates of abnormal pulmonary pressure, resistance, and compliance, and their association with incident heart failure (HF) in a subset of 3,154 elderly individuals (76 years old) free of HF who underwent echocardiography at the 5th study visit (2011-2013) and had measurable tricuspid regurgitation by spectral Doppler. Echocardiography-derived pulmonary artery systolic pressure (PASP), resistance (PVR), and compliance (PAC) were measured. 

The authors found that mean values of PASP, PVR and PAC were 28±6 mmHg, 1.7±0.4 WU and 3.4±1.0 mL/mmHg respectively, and were abnormal in 19%, 13% and 15%, respectively (compared to 10th and 90th percentile limits in 253 low risk participants free of cardiovascular disease or risk factors. 

PASP, PVR, and PAC were each predictive of incident HF or death (all adjusted p <0.001) independent of LV measures. Pulmonary hypertension was a strong predictive factor of incident HFpEF (HFpEF: 2.5 [1.6-3.8], HFrEF: 1.8 [1.1-2.9]) while abnormal PAC more strongly

predicted incident HFrEF (HFpEF: 1.6 [0.9-2.8], HFrEF: 2.3 [1.3-4.2]). Abnormal PVR was not predictive of either of HF outcomes (HFpEF: 1.0 [0.6-1.9], HFrEF: 0.6 [0.3-1.1]). The authors concluded that impairments in pulmonary vascular function may antecede clinical HF, and that a comprehensive pulmonary hemodynamic evaluation may identify pulmonary vascular phenotypes that differentially predict HF phenotypes.

This is a pertinent, nice-written paper, performed in a large sample of elderlies without known clinical HF, in which authors suggest that impairments in pulmonary vascular may have a role identifying patients at higher risk of HF and death. There are some issues to discuss.

1. Age-appropriate reference limits for pulmonary vascular measures were defined in a subset of low risk participants free of prevalent cardiovascular disease or risk factors. In my opinion, it is crucial to explain why this subset was defined as reference. Why not to compare with other established cut-off values?

2. Clarify how left heart dysfunction was defined. It was present in 30% at baseline. Patients with LVEF<40% at baseline were excluded from the study? 

3. Cox proportional hazards models were employed to assess the association of PASP, PVR and PAC with incident HF overall, incident HFrEF incident HFpEF, and the composite (HF/death). Please, clarify the length of follow-up, the number of events (persons/years) and the covariates include in all models.

4. The study sample had no prior HF (theoretically). However, this is a high-risk population for HF (76 years old, 82% prior hypertension, and 34% type 2 DM). I speculate a representative proportion of these patients may have unrecognized clinical HF at baseline. What was the distribution of NT-proBNP at baseline? This is a crucial point. Clinical HF was ruled out? How?

4. Along the same line, the authors showed a significant association between pulmonary vascular dysfunction (PASP, PVR, and PAC) with NT-proBNP at baseline. I speculate some of these patients may have clinical HF at baseline. How many of these patients fulfilled the criteria for diagnosing HF at baseline. Please, report. It should be also interesting to include NT-proBNP in all models, especially in those prognostic ones and the evaluate the prognostic ability of parameters of pulmonary dysfunction across the levels of NT-proBNP. This is issue should also be discussed more in depth. 

5. The article is difficult to follow. There is a too much data/comparisons. 

Reviewer #3: In this study by Santos et al, investigators examined the association echocardiography-derived pulmonary hemodynamic measures (PASP, PVR, and PAC) with incident HF among 3,154 ARIC participants. Abnormal PASP was associated with incident HF including both HFpEF and HFrEF, whereas abnormal PVR was not associated with HF outcomes. This is a well-written manuscript, and analyses are well done. The paper demonstrates that pulmonary vascular function ascertained by echocardiography is correlated with both left heart and lung dysfunction, and associates with greater risk of future heart failure. 

Specific comments: 

- Methods: authors should consider addition of well-established risk factors for HF and PH to multivariable-adjusted models, including BMI, DM, HTN, which were not included - these may act as potential confounders given their associations with both PH and HF. 

- Results: Authors define abnormal PASP, PVR, and PAC cut-points using reference limits. It is important to acknowledge that these reference limits represent lower cut-points when compared with accepted clinical measures. For example, PASP >=40 is usually what prompts further evaluation for PH (vs PASP deemed abnormal at >=32 here). With this definition of abnormal, it appears that more than half of individuals deemed to have "abnormal PVR" do not have abnormal PASP (Figure 1A) - is mean PAP elevated in those cases or how can abnormal PVR in absence of PAP elevation be understood? 

- Results: In S11 Table, none of the measures (PASP, PVR, PAC, when analyzed as continuous variables) were associated with HFrEF - yet when presented as dichotomous variables, PASP and PAC are associated. Can authors comment on this discrepancy? Is there a non-constant HR across the range of PASP? 

- Differential prediction of HFpEF vs HFrEF: "abnormal PASP was predictive of both incident HFpEF and HFrEF, with a higher magnitude of effect noted for incident HFpEF" - did authors formally test for difference in HR? The number of outcomes here are small and I would encourage authors not to place too much emphasis on differential associations with HF subtypes in both the abstract and discussion

- Discussion: authors suggest that pulmonary hypertension leads to development of HFpEF / HF - could it be that HF was just undiagnosed at the time of PV assessment and the two co-exist or PH is a manifestation of HFpEF (how much time between echo and HF diagnosis)? How many individuals may meet echo/NT-proBNP criteria for HFpEF at the time of echo? Would be careful with wording - authors did not diagnose pulmonary hypertension - the cut-points used in this paper are derived reference limits but are lower compared with clinical definitions of PH. 

Reviewer #4: This manuscript investigates the prevalence, correlates and outcomes associated with several indexes of pulmonary vascular dysfunction (PASP, PVR, PAC) in the well known ARIC cohort. 

There were the expected associations with left heart and pulmonary dysfunction. 

Additional to PASP, PVR and PAC were each predictive of incident HF or death, independent of LV measures, suggesting some utility in obtaining these other parameters.

The most intriguing finding was that PASP more strongly predicted incident HFpEF than HFrEF, while abnormal PAC predicted future HFrEF but not HFpEF. Abnormal PVR did not predict either HF phenotype.

These main messages are clearly communicated. 

This work would be an important contribution to the literature. 

Major strengths of the study include 

- well conceived and executed methodology

- large established cohort with dense phenotyping

- more comprehensive noninvasive characterization of pulmonary vascular function in a large community-dwelling cohort, which is unique

- biologically plausible findings.

My comments and concerns follow:

1) A large majority of the study participants were female.

While this is a corrective to the often male-dominated clinical studies, it is not reflective of the general US and other populations worldwide.

Conclusions drawn should be cautiously interpreted.

2) Why was a single measure of RAP used when determining PASP?

This is not in keeping with previous and current ASE guidelines.

3) Please provide more information on HFpEF outcomes.

- how was LVEF adjudicated in these cases?

- were 2016 ESC criteria for HFpEF satisfied in the majority of cases? 

4) The authors tout the value of characterizing multiple measures of pulmonary vascular function, other than PASP.

They should highlight also the limitations of noninvasive determination of PAC and PVC.

PAC is indirectly derived from PASP, as well as other measures that can be technically challenging to accurately ascertain.

PVR is also dependent on TR velocity while RVOT outflow velocity curve is not infrequently technically inadequate. 

This may be why the number of subjects with measurable PAC and PVR (detailed in see Fig 3) are lower, and in the case of PAC, much lower than subjects with measurable PASP.

While this is a large study, the smaller number of outcomes arising from the reduced sample size with measurable PAC does increase the potential for statistical over-fitting.

5) Did PAC, when analyzed dichotomously, have incremental value in predicting HFpEF over PASP?

With respect to outcomes summarized graphically in Fig 3, was PAC still predictive of HFpEF after correction for PASP?

6) Line 397: "Both high PVR and lower PAC were associated with worse RV systolic function, while PASP - within the range represented in this study sample - was not associated with RVFAC."

The authors will be well aware that PASP is dependent on RV function, and by itself does not provide information on right ventricular-pulmonary arterial coupling. 

RV-PA coupling has been shown in several publications to predict death and recurrent HF in subjects with HF, and the risk of RV failure in patients with pulmonary hypertension.

Since the authors have data on RV function, i.e. fractional area change and tricuspid annulus systolic velocity, it would be instructive to evaluate if a measure of RV-PA coupling would be even more useful in predicting adverse HF and other outcomes.

Minor concerns

- Line 321: "...emerging as an important contributor to HF general..."

Should read "HF in general"

- Line 336: "...abnormalities of distinct dimensions of pulmonary vascular function..." 

What does this mean?

- Line 351: "(RC time)"

Please qualify all abbreviations.

[LINK]

---

## [Decision Letter · Decision Letter 1]

11 Aug 2020

Dear Dr. Shah,

Thank you very much for re-submitting your manuscript "Pulmonary vascular dysfunction among people aged over 65 year in the community in the Atherosclerosis Risk In Communities (ARIC) study: a cross-sectional analysis" (PMEDICINE-D-19-04388R1) for review by PLOS Medicine.

I have discussed the paper with my colleagues and the academic editor and it was also seen again by reviewers. I am pleased to say that provided the remaining editorial and production issues are dealt with we are planning to accept the paper for publication in the journal.

[LINK]

We look forward to receiving the revised manuscript by Aug 18 2020 11:59PM. 

Sincerely,

Adya Misra, PhD

Senior Editor 

PLOS Medicine

plosmedicine.org

Requests from Editors:

Please provide full stops after reference brackets 

Line 469 should say “heightened” perhaps

Data- please provide an accession number or direct link to the dataset. If there are restrictions on data sharing, please mention these in the data availability statement, providing reasons for these

. 

Author summary-could you reword “time to event analysis” using more accessible language? Please also remove "Non technical" from the author summary heading

Please provide the STROBE checklist as a separate supplementary file and include a callout to this file in the methods. Please also use paragraphs and sections instead of page numbers as these are likely to change. 

Please provide a direct link to the interviewer administered questionnaires used in ARIC and cite this where the questionnaires are mentioned in the methods section

Comments from Reviewers:

Reviewer #1: Thank you to the authors for addressing my previous comments well.

Reviewer #2: This is a very interesting topic suggesting that impairment in pulmonary vascular function may precede clinical heart failure. In this revised version of the manuscript, the authors have succesfully answered all the issues raised in my prior revision.

Reviewer #3: The authors have comprehensively responded to comments - I especially appreciate additional analyses to exclude anybody with possible prevalent HF, as well as tables reassuring that findings are similar when restricting to clinically used cut-points for PASP and PVR. These additional analyses strengthen the primary findings. 

Reviewer #4: This is a revised manuscript which I previously reviewed.

Comments of the editors and all other reviewers were noted.

I believe that the majority of concerns raised have been adequately addressed.

I have some additional comments on the revisions made:

1) Methods section, 'Assessment of Pulmonary Hemodynamics' sub-section, Line 166-168:

For reproducibility metrics of pulmonary vascular variables, coefficients of variation and correlations are not meaningful.

Please provide the intraclass correlation coefficients for the raw measurements.

Results section, "Associations of left heart disease, pulmonary dysfunction, and venous thromboembolism with pulmonary vasculature dysfunction" sub-section, Line 320-324 and 

Results section, "Relationship of pulmonary vascular hemodynamics with NT-proBNP, RV function, and incident HF" sub-section, Line 375-381:

Why were only those 196 participants with moderate or severe dyspnea and ESC HFA criteria for HFpEF excluded? 

Why not also those with LVEF <50% using the 2016 ESC criteria for HFrEF and HFmrEF?

Results section, 'Relationship of pulmonary vascular hemodynamics with NT-proBNP, RV function, and incident HF' sub-section, Line 362-369:

"...mean±SD follow-up of 5.3±1.2 years within range of 6.6 years..." does not look right.

Table S11 appears to be truncated - there are no data on PAC.

Minor concerns:

Response/Supplementary Tables:

- Even rate should be Event rate

- PAR (95%) should be PAR (95% CI)

There are several spelling errors.

[LINK]

---

## [Editor Report · Decision Letter 2]

31 Aug 2020

Dear Dr. Shah, 

On behalf of my colleagues and the academic editor, Dr. Kazem Rahimi, I am delighted to inform you that your manuscript entitled "Pulmonary vascular dysfunction among people aged over 65 year in the community in the Atherosclerosis Risk In Communities (ARIC) study: a cross-sectional analysis" (PMEDICINE-D-19-04388R2) has been accepted for publication in PLOS Medicine. 

PRODUCTION PROCESS

PRESS

PROFILE INFORMATION

Thank you again for submitting the manuscript to PLOS Medicine. We look forward to publishing it. 

Best wishes, 

Adya Misra, PhD

Senior Editor 

PLOS Medicine

plosmedicine.org